# The Antioxidant Properties of *Lavandula multifida* Extract Contribute to Its Beneficial Effects in High-Fat Diet-Induced Obesity in Mice

**DOI:** 10.3390/antiox12040832

**Published:** 2023-03-29

**Authors:** Jose Alberto Molina-Tijeras, Antonio Jesús Ruiz-Malagón, Laura Hidalgo-García, Patricia Diez-Echave, María Jesús Rodríguez-Sojo, María de la Luz Cádiz-Gurrea, Antonio Segura-Carretero, José Pérez del Palacio, María Reyes González-Tejero, María Elena Rodríguez-Cabezas, Julio Gálvez, Alba Rodríguez-Nogales, Teresa Vezza, Francesca Algieri

**Affiliations:** 1Department of Pharmacology, Center for Biomedical Research (CIBM), University of Granada, 18071 Granada, Spain; 2Instituto de Investigación Biosanitaria de Granada (ibs. GRANADA), 18012 Granada, Spain; 3Department of Analytical Chemistry, Faculty of Science, University of Granada, 18071 Granada, Spain; 4Fundación MEDINA, Centro de Excelencia en Investigación de Medicamentos Innovadores en Andalucía, 18016 Granada, Spain; 5Department of Botany, University of Granada, 18071 Granada, Spain; 6Centro de Investigación Biomédica en Red de Enfermedades Hepáticas y Digestivas (CIBEREHD), Instituto Salud Carlos III, 28029 Madrid, Spain; 7Servicio de Digestivo, Hospital Universitario Virgen de las Nieves, 18014 Granada, Spain

**Keywords:** obesity, high-fat diet, insulin resistance, inflammation, *Lavandula multifida*

## Abstract

Obesity is a worldwide public health problem whose prevalence rate has increased steadily over the last few years. Therefore, it is urgent to improve the management of obesity and its comorbidities, and plant-based treatments are receiving increasing attention worldwide. In this regard, the present study aimed to investigate a well-characterized extract of *Lavandula multifida* (LME) in an experimental model of obesity in mice and explore the underlying mechanisms. Interestingly, the daily administration of LME reduced weight gain as well as improved insulin sensitivity and glucose tolerance. Additionally, LME ameliorated the inflammatory state in both liver and adipose tissue by decreasing the expression of various proinflammatory mediators (*Il-6*, *Tnf-α*, *Il-1β*, *Jnk-1*, *Pparα*, *Pparγ*, and *Ampk*) and prevented increased gut permeability by regulating the expression of mucins (*Muc-1*, *Muc-2*, and *Muc-3*) and proteins implicated in epithelial barrier integrity maintenance (*Ocln*, *Tjp1*, and *Tff-3*). In addition, LME showed the ability to reduce oxidative stress by inhibiting nitrite production on macrophages and lipid peroxidation. These results suggest that LME may represent a promising complementary approach for the management of obesity and its comorbidities.

## 1. Introduction

Obesity is recognized as one of the most prevalent worldwide public health problems of the modern era. According to the World Health Organization, it is an ongoing epidemic that knows no borders and keeps growing, despite the policy interventions implemented in the last few years. It is estimated that 2.7 billion adults will be overweight, and over 1 billion obese by 2025 [1]. Obesity is a chronic disease with a complex etiopathogenesis involving genetic, biological, behavioral, psychological, cultural, and socioeconomic factors [2]. It is driven by a continual positive energy balance that gives rise to excessive body fat accumulation. Moreover, obesity increases the risk of chronic diseases such as cardiovascular diseases, type 2 diabetes, and different types of cancer that can worsen the quality of life and reduce life expectancy [3,4,5].

Different anti-obesity drugs are available on the market, but most of them show important side effects and lack long-term efficacy [6]. This limitation has led to the search for alternative and/or complementary approaches for treating and preventing this disorder. Herbal remedies and functional foods of plant origin have been the basis of traditional medicine and are commonly used by obese patients [7]. One of the plant families most often applied in traditional medicine is *Lamiaceae*, which is very abundant in the degraded areas of the Mediterranean maquis and rocky, calcareous, or sandy soils. The genus *Lavandula* (*lavender*) includes 39 species that are traditionally employed to treat different diseases, including diabetes [8,9], inflammatory conditions, and digestive complaints [10,11]. It has been described that these crude drugs are rich in a wide range of secondary metabolites that are responsible for their beneficial properties, including antimicrobial, antifungal, and antioxidant activities [9]. Among these *Lavandula* species, *L*. *multifida* has been traditionally used against rheumatism and other inflammatory diseases [10], as well as for the treatment of diabetes and infectious diseases [12]. This study intended to assess the effects of a characterized hydroalcoholic extract of *L. multifida* (LME) on an experimental model of diet-induced obesity in mice, paying special attention to its impact on the altered metabolic and inflammatory status. Furthermore, the antioxidant and anti-adipogenic properties of the extract were also assessed in vitro in RAW 264 macrophages and 3T3-L1 MBX preadipocytes, respectively.

## 2. Materials and Methods

### 2.1. Chemicals and Reagents

The analytical procedures were carried out by employing water purified by a Milli-Q system from Millipore (Bedford, MA, USA), LC-MS grade acetonitrile, and acetic acid, which were purchased from Fisher Chemicals (Waltham, MA, USA), and Sigma-Aldrich (Steinheim, Germany), respectively.

The following reagents were provided from the specified suppliers: sodium carbonate, acetic acid, TPTZ (2,4,6-tris(2-pyridyl)-s-triazine), sodium hydroxide, and hydrochloric acid were acquired from Fluka (Honeywell, NC, USA). Absolute ethanol was purchased from Riedel-de-Haën (Honeywell, NC, USA). Gallic acid, Folin reagent, ABTS (2,2′-azinobis (3-ethylbenzothiazoline-6-sulphonate)), potassium persulfate, Trolox (6-hydroxy-2,5,7,8-tetramethylchroman-2-carboxylic acid), sodium acetate, ferric chloride, heptahydrate ferrous sulfate, fluorescein, AAPH (2,2′-azobis(2-amidinopropane) dihydrochloride), sodium phosphate monobasic, and dibasic were obtained from Sigma-Aldrich (St. Louis, MO, USA). 

All chemicals, unless otherwise indicated, were purchased from Sigma-Aldrich (Merck Life Science S.L.U., Madrid, Spain).

### 2.2. Plant Material and Preparation of the Extract

The aerial parts of *L. multifida* were collected in Ízbor (36°53′86″ N 3°30.08′25″ W; Granada, Spain) in April 2015. The plant was identified and authenticated by Dr. M. R. González-Tejero and Dr. J.A. Hita from the department of Botany of the University of Granada, Spain. The voucher specimens corresponding to *L. multifida L.* (GDA 62621) were deposited in the herbarium of the University of Granada (Granada, Spain). The plant extracts were prepared as described before [13]. Briefly, 5 g of ground plant material was mixed with washed sea sand (Panreac Química S.A.U., Castellar del Vallès (Barcelona, Spain) and extracted with 30 mL of methanol 50% (*v*/*v*) at 1500 PSI and 80 °C for 10 min in an ASE200 extraction system (Dionex Corporation, Sunnyvale, CA, USA). After two extraction cycles, liquid extracts were pooled, and the solvent evaporated under a vacuum at 60 °C. The extraction efficiency for *L. multifida* (expressed as a percentage *w*/*w*) was 17%.

### 2.3. Chemical Composition of LME Using UHPLC-MS Conditions

The qualitative characterization of LME was carried out using a specifically ACQUITY UPLC H-Class System (Waters, Milford, MA, USA) coupled to an electrospray quadrupole-time of flight mass spectrometer (ESI-qTOF-MS, Synapt G2, Waters Corp., Milford, MA, USA), working in negative-ion mode over a range from 50 to 1200 *m*/*z* and following the literature [14] with minor changes. Briefly, the dried extract was redissolved to a concentration of 5 mg/mL and filtered through a 0.2 μm filter before analysis. The separation was carried out in an ACQUITY UPLC BEH Shield RP18 Column, 130 Å, 1.7 µm, 2.1 mm × 150 mm. The injection volume was 10 µL, and the phytochemical separation was performed at room temperature according to this multistep gradient: 0.0 min 99% A; 2.33 min 99% A; 4.37 min 93% A; 8.11 min 86% A; 12.19 min 76% A; 15.99 min 60% A; 18.31 min 2% A; 21.03 min 2% A; 22.39 min 99% A; and 25.0 min 99% A. The mobile phases were acidified water (0.5% acetic acid, *v*/*v*) as solvent A and acetonitrile as solvent B. Finally, the total flow rate was fixed at 0.7 mL/min. MS acquisition was performed using 2 parallel scan functions by rapid switching, in which 1 scan was operated at low collision energy in the gas cell (4 eV) and the other at high collision energy (MSE energy linear ramp: from 20 to 60 eV); desolvation gas flow = 700 L/h, desolvation temperature = 500 °C, cone gas flow = 50 L/h, source temperature = 100 °C, capillary voltage = 2.2 kV, cone voltage = 30 V, and collision energy = 20 eV. The scan duration was 0.1 s, and the resolution was 20,000 FWHM. The MS data were managed with the open-source software MZmine 2.53.

### 2.4. Total Phenolic Content and Antioxidant Capacity Assays

Folin–Ciocalteu (TPC), FRAP, TEAC, and ORAC were performed on a Synergy H1 Monochromator-Based Multi-Mode Microplate reader (Bio-Tek Instruments Inc., Winooski, VT, USA) following the methodology previously reported [15]. All measurements were made in triplicate.

The 1,1-diphenyl-2-picrylhydrazyl (DPPH) test was made to evaluate the free radical scavenging activity of LME [15]. Briefly, LME, gallic acid, epicatechin, and the positive control ascorbic acid were dissolved in methanol to reach a range of concentrations of 0.1–100 µg/mL. Then, 10 µL of each methanolic dilution was mixed with 90 µL of phosphate buffer at pH = 7 and a 200 µL of a DPPH solution at 100 µM. The plate was protected from light and incubated for 30 min at 25 °C. After incubation, the scavenging activity of each compound was measured at 515 nm in a Magellan^®^ Tecan Infinite F50 spectrophotometer (Tecan Group Ltd., Männedorf, Switzerland). The percentage of radical DPPH scavenging activity (% RSA) was calculated for each concentration employing the following equation:RSA (%) = [(Abs Blank − Abs Sample)/Abs Blank] × 100
where Abs Blank and Abs Sample are the absorbance values at 515 nm of the blank and samples, respectively. The blank absorbance value is the maximum and corresponds with the highest levels of the DPPH radical. Then, the half-maximal inhibitory concentration (IC50) value was calculated for each compound tested. 

### 2.5. In Vitro Studies

RAW 264 cells (a mouse macrophage cell line) and 3T3-L1 MBX cells (a mouse preadipocyte cell line) were obtained from the Cell Culture Unit at the University of Granada (Granada, Spain). Cells were grown in Dulbecco’s Modified Eagle Medium (DMEM) (Gibco, ThermoFisher Scientific, Waltham, MA, USA) supplemented with a 10% heat-inactivated fetal bovine serum (FBS) for RAW 264 cells or heat-inactivated fetal calf serum (FCS) for 3T3-L1 MBX cells, L-glutamine (2 mmol/L), penicillin (100 units/mL), and streptomycin (100 units/mL) in a humidified 5% CO_2_ atmosphere at 37 °C. 

RAW 264 cells were seeded into 96-well plates at a density of 5 × 10^5^ cells/well and grown until the formation of a monolayer. Then, they were pre-incubated with different concentrations of LME ranging from 0.1 to 100 μg/mL for 2 h and stimulated with the lipopolysaccharide (LPS) from *Escherichia coli* 055:B5 (100 ng/mL) for 24 h. For negative and positive controls, untreated unstimulated and LPS-stimulated cells were used, respectively. After the stimulation period, the supernatants were harvested for nitrite determination by the Griess Assay [16]. An amount of 100 μL of cell supernatant were mixed with 100 μL of Griess reagent (0.1% N-(1-naphthy) ethylenediamine solution and 1% sulphanilamide in 5% (*v*/*v*) phosphoric acid solution, mixed in a proportion 1:1) and incubated for 10 min. A colored azolic compound was formed and its concentration was measured by a photometric measurement of the absorbance at 550 nm. Cell viability of tested conditions was evaluated by the MTS-based CellTiter 96^®^ AQueous One Solution Cell Proliferation Assay (Promega, Madison, WI, USA), being the cellular viability estimated from the absorbance value and by comparing it with the untreated control cells.

3T3-L1 MBX cells were seeded into 6-well plates at 5 × 10^6^ cells/well density. 2 days soon after cells reached confluence (day 0), different prodifferentiative agents (0.5 mM 3-isobutyl-1-methylxanthine (IBMX), 5 μM dexamethasone, and 10 μg/mL insulin) were added to the medium to initiate cell differentiation. After 48 h (day 2), the culture medium was changed to DMEM and 10% fetal bovine serum (FBS) contained only insulin (10 μg/mL). On day 4, the medium was replaced with DMEM and 10% FBS, and the cells were incubated for an additional 2 days until fully differentiated adipocyte-like cells were obtained. Treatment of 3T3-L1 cells with LME (10, 25, and 100 μg/mL) was initiated on day 0 and added whenever the medium was replaced. Images from 3T3-L1 cells were taken to visualize adipocyte differentiation on days 2, 4, and 6.

### 2.6. Animals and Experimental Design

The study was carried out following the “Guide of the Care and Use of Laboratory animals” as promulgated by the National Institute of Health, and all procedures were approved by the Ethics Committee of Laboratory Animals at the University of Granada (Spain) (Ref. No. 28/03/2016/030). A total of 8-week-old male C57BL/6 mice acquired from Janvier labs (St. Berthevin, Cedex, France) were housed in Makrolon cages, kept under controlled light-dark cycles (12 h light/dark), temperature and relative humidity (22 ± 1 °C, 55 ± 10%), and with free access to tap water. They were randomly assigned to 3 groups (*n* = 8): control diet (CD), high-fat diet (HFD), and HFD-treated group with LME (25 mg/kg) dissolved in water by gavage during all the experimental procedures. Standard chow diet (13% calories from fat, 20% calories from protein, and 67% calories from carbohydrate; Global diet 2014) and high-fat diet (59% calories from fat, 13% calories from protein, and 28% calories from carbohydrate; Purified diet 230 HF) were purchased from Harlan Laboratories (Barcelona, Spain) and Scientific Animal Food and Engineering (Augy, France). Mice were fed the diets and treated with the extract for 30 days. Animal body weight and food and water intake were monitored, and energy efficiency was estimated as the ratio of weight gain (g) to caloric intake (Kcal). Mice were sacrificed by cervical dislocation, and liver and abdominal and epididymal fat were collected, cleaned, and weighed. Fat/weight index was calculated by dividing body weight by tibia length. The samples were frozen in liquid nitrogen and stored at −80 °C until additional analysis.

### 2.7. Glucose Tolerance Test

During 1 week before the sacrifice, mice fasted for 8 h and were given a 2 g/kg of body weight glucose solution by intraperitoneal injection. Blood was sampled from the tail vein at 0, 15, 30, 60, and 120 min after injection. A handheld glucometer (Contour XT, Ascensia Diabetes Care, S.L., Barcelona, Spain) was used to determine glucose levels. 

### 2.8. Biochemical Determinations

Before sacrifice, mice fasted overnight, and a blood sample was taken by cardiac puncture under isoflurane anesthesia in heparin blood collection tubes. Blood samples were centrifuged for 20 min at 5000× *g* at 4 °C, and the resulting plasma was frozen at −80 °C until further analysis. Plasma glucose, LDL (low-density lipoprotein)-cholesterol, and HDL (high-density lipoprotein)-cholesterol concentrations were determined by colorimetric methods using Spinreact kits (Spinreact, S.A., Girona, Spain). Plasma insulin levels were measured using a mouse insulin ELISA Kit (Alpco Diagnosis, Salem, NH, USA). A homeostatic model assessment of insulin resistance (HOMA-IR) was computed with the formula: fasting glucose (mM) × fasting insulin (μU/mL)/22.5. LPS plasma levels were quantified using a Pierce™ Chromogenic Endotoxin Quant Kit (Thermo Scientific, Inc., Waltham, MA, USA) according to the manufacturer’s recommendations. 

### 2.9. Thiobarbituric Acid Reactive Substance Assay

Mouse liver was removed, lysed, and homogenized, and the protein concentration was calculated using the colorimetric method of Bicinchoninic Acid Assay (BCA). Then, the lipid oxidation was determined in the samples by measuring the amount of thiobarbituric acid reactive substances (TBARS), as described before [17]. Briefly, the malondialdehyde (MDA), resulting from lipid peroxidation, reacts with the thiobarbituric acid (TBA) used in the extraction method. The products of the reaction are TBARS, and its absorbance is assessed at 535 nm. TBARS levels were expressed as µM/mg protein in liver tissue.

### 2.10. Histological Studies

Samples of epididymal adipose tissue and liver were fixed in 4% PFA, dehydrated, and embedded in paraffin. Then, 5 µm-thick sections were cut and stained with hematoxylin and eosin. Adipocyte size was estimated and evaluated using the Fiji imaging software with the Adiposoft v1.16 plugin.

### 2.11. Analysis of Gene Expression by RT-qPCR 

Total RNA was extracted from adipose tissue and liver samples with the RNeasy Mini Kit (Qiagen, Germantown, MD, USA), following the recommended protocol. RNA quantification was obtained using a NanoDrop 200c spectrophotometer (Life Technologies, Thermo Fisher Scientific), and purity was confirmed with the 260 nm/280 nm and 260 nm/230 nm absorbance ratios. Subsequently, RNA was reverse transcribed into cDNA using oligo(dT) primers (Promega, Southampton, UK). Real-time quantitative PCR (qPCR) amplification and detection were carried out on optical-grade 48 well plates in the EcoTM Real-time PCR System (Illumina, San Diego, CA, USA) using 20 ng of cDNA, the MasterMix qPCR SyGreen Kit (PCR Biosystems Ltd., London, UK), and the specific primers (at a final concentration of 100 nM) Appendix A. The mRNA relative quantitation was estimated with the ΔΔCt method and glyceraldehyde 3-phosphate dehydrogenase (*Gapdh*) was employed as a housekeeping gene.

### 2.12. Analysis of Protein Expression by Western Blot 

Proteins were separated in a 10% SDS-PAGE and then transferred to a PVDF membrane (GE Healthcare Life Sciences, Marlborough, MA, USA). They were blocked and probed at 4 °C overnight with the following anti-mouse antibodies: anti-AMPK (1:2000 dilution), anti-p-AMPK (1:1000 dilution), anti-AKT (1:1000 dilution), anti-p-AKT (1:1000 dilution), and anti-PPARγ (1:1000 dilution) (Cell Signaling, Danvers, MA, USA) or anti-SIRT1 (1:1000 dilution). This was followed by 1 h of incubation with peroxidase-conjugated anti-rabbit IgG antibody (1:5000 dilution) and β-actin (1:1000 dilution) (Santa Cruz Biotechnology, Inc., Heidelberg, Germany). The specific proteins were identified by Western Lightning™ Chemiluminescence Reagent Plus (PerkinElmer Spain SL, Madrid, Spain) and semi-quantified by the ImageJ software (Free Software Foundation Inc., Boston, MA, USA). 

### 2.13. Statistic

All results are expressed as the mean ± SEM. Differences between means were assessed for statistical significance using a one-way analysis of variance (ANOVA) and post hoc least significance tests. Differences between proportions were analyzed with the chi-squared test. All statistical analyses were performed with the GraphPad 8 software package (GraphPad Software, Inc., La Jolla, CA, USA), with statistical significance set at *p*-value < 0.05.

## 3. Results and Discussion

Obesity is a chronic, often progressive condition recognized as an escalating risk in the development of metabolic alterations, including dyslipidemia and glucose intolerance, or cardiovascular diseases, such as hypertension or atherosclerosis [18,19]. At present, the approach to obesity implies the establishment of important modifications in lifestyle, such as caloric restriction or physical exercise, which are frequently difficult to maintain with time. Moreover, the pharmacological treatment of obesity is now plausible, with the administration of lipase inhibitors or anorexigenic drugs; however, they usually show limited efficacy and important side effects [6]. In this context, and considering the relevance of obesity and its co-morbidities in human health, more efficient and safer treatments are required for its management. This could be the case with plant extracts used traditionally for different purposes. For instance, LME contains different active compounds, including polyphenols, which can contribute to its beneficial effects, given the well-known antioxidant and anti-inflammatory properties ascribed to these compounds [20].

### 3.1. Chemical Characterization of LME

The present study has characterized the chemical composition of LME by UHPLC-MS for the first time. Figure 1 shows the base peak chromatogram (BPC) of LME. A total of 65 compounds were identified, primarily as pentacyclic triterpenes.

They are included in Table 1 and numbered in accordance with their elution order. Moreover, this table specifies their retention times (RT), experimental *m*/*z*, molecular formula, and proposed compounds.

### 3.2. Antioxidant Capacity of LME

As a previous step to further validate the antioxidant potential of this extract by ORAC, TEAC, and FRAP assays, the total phenolic content (TPC) was estimated in LME by the Folin–Ciocalteu method, which was 179 mg GAE/g of plant extract. Table 2 displays the values obtained for every assay. Based on the results obtained, LME showed free radical scavenging activity (Table 2). 

The antioxidant activity of LME was also evaluated by carrying out the DPPH assay, a widely used method based on the reduction of alcoholic DPPH solution in the presence of hydrogen-donating antioxidants. LME displayed a dose-dependent neutralizing activity of 10.4%, 21.2%, and 58.6%, at concentrations of 0.1, 1, and 10 µg/mL of extract, respectively (Figure 2A). The free radical scavenging activity of LME (IC50 = 8.06 µg/mL) was compared with that of some compounds with a well-known ability to neutralize DPPH radicals: gallic acid (IC50 = 7.94 µg/mL), epicatechin (IC50 = 6.43 µg/mL) and ascorbic acid (IC50 = 7.91 µg/mL) [21,22]. 

Interestingly, numerous studies have reported that the intake of an HFD can cause oxidative stress and increase lipid peroxidation, thus releasing various reactive aldehydes such as 4-hydroxynonenal (HNE) and MDA [23,24]. In addition to being considered biomarkers of lipid peroxidation, these compounds can cause DNA damage, thus inducing pathological processes such as cytotoxicity [24]. As expected, lipid peroxidation was higher in those mice fed HFD resulting in a significant increase of TBARS compared to CD-fed mice. The antioxidant activity exhibited by LME in vitro was also evidenced in vivo, in which a significant reduction of the amount of TBARS in treated HFD mice (Figure 2B) was observed. Phenolic compounds such as rutin, quercetin glucoside, and epicatechin gallate may be responsible for the restoration of the antioxidant status in mice treated with LME.

### 3.3. Effects of LME on Nitrite Production in RAW 264 Cells and Adipogenesis in 3T3-L1 Cells

As commented above, oxidative stress is an important pathogenic mechanism of obesity and its associated complications [25]. Excessive weight gain is characterized by increased adipocyte size and macrophage recruitment, in association with increased free radical and reactive oxygen species production, which contribute to the establishment of a chronic inflammatory state and, consequently, metabolic dysfunctions [26]. Specifically, pro-inflammatory macrophages contribute to this oxidative stress status by the induction of different enzymes, including inducible NO synthase (iNOS), whose induction favors the release of nitric oxide (NO), a key regulator of body composition and energy metabolism [27], thus influencing both adipogenesis and insulin resistance. 

The effects of LME were evaluated in vitro in murine RAW 264 macrophages. The incubation of RAW 264 cells with different concentrations of LME (0.1–100 μg/mL) during 24 h did not show NO accumulation. Similarly, cell viability was not significantly modified by any of the doses tested (Figure 3). However, LPS (a potent activator of inflammatory signaling pathways) induced a marked NO production, which was dose-dependently decreased by LME pretreatment (Figure 3). 

In addition, abnormal accumulation of lipids in cells is a characteristic of insulin resistance and obesity. Thus, the reduction of elevated cellular lipid levels could be a potential approach for the management of these pathological conditions. In this sense, the impact of various concentrations of LME on lipid accumulation was assayed in pre-adipocyte 3T3-L1 cells, which was checked throughout the experiment. As expected, under appropriate differentiation conditions, 3T3-L1 mature cells showed many lipid droplets in comparison to pre-adipocytes (Figure 4A). Interestingly, LME at the highest dose (100 μg/mL) revealed a lipid-lowering effect, thus highlighting its regulatory impact on adipogenesis. Since adipogenesis is regulated by various transcription factors and adipogenesis-related genes, we continued evaluating LME impact on the protein level of transcriptional factor PPARγ by western blot. PPARγ is a critical component in adipogenesis, where its over-expression aggravates the intracellular triglycerides accumulation in adipocytes as well as cell size. As shown in Figure 4B, the PPARγ level was slightly diminished by LME, especially at the dose of 100 μg/mL. Thus, lower intracellular lipids accumulation observed with LME could be linked to direct changes in the PPARγ protein level. Nevertheless, the exact underlying mechanisms remain to be elucidated. It could be associated with the regulation of other transcriptional factors such as, i.e., CCAAT/enhancer-binding proteins alpha and beta or sterol-regulatory element binding protein-1c, although this has not been explored yet.

### 3.4. Effects of LME on Weight Evolution, Glucose Tolerance Test, and Plasma Biochemical Profile

HFD consumption increased body weight gain over 31 days compared to those mice that were fed a control diet (Figure 5A), according to previous studies [28,29]. However, the daily administration of LME to HFD-fed mice significantly diminished weight gain from day 6 (Figure 5A); of note, no satiating effect was shown since similar food-intake values were observed in all HFD-fed groups throughout the experiment (Figure 5B,C). Similarly, the extract ameliorated the glucose metabolism impairment observed in control HFD mice, as evidenced in the glucose tolerance test, which was completed one week before the end of the study. Thus, lean mice displayed a peak in blood glucose levels approximately 15 min after i.p. glucose administration (2 g/kg), followed by a return to baseline values approximately 60 min after the glucose challenge, suggesting a proper glucose metabolism (Figure 5D). Untreated HFD-fed mice exhibited significantly higher plasma glucose concentrations than CD mice at all the time points evaluated, while LME treatment markedly decreased these glucose levels from 15 min onwards, thus causing reductions in the area under the curve (AUC) (Figure 5D). Likewise, obese mice receiving LME displayed a significant decrease in the plasmatic glucose levels, thus supporting the amelioration in the glucose intolerance status observed with the glucose tolerance test. Moreover, although no difference in fasting plasma insulin level was observed, LME was able to reduce insulin resistance indicated by the HOMA-IR index (Figure 5E). 

HFD consumption also produced a dyslipidemia status in control obese mice, characterized by elevated serum levels of total cholesterol and triglycerides compared to CD-fed mice, being the extract able to significantly reverse this plasma lipid profile impairment (Figure 5E).

The alterations in fatty acid metabolism associated with obesity were also corroborated by histological analysis. Liver sections stained with hematoxylin and eosin revealed important steatosis in the untreated obese group, mainly characterized by intense fat deposition within the cytoplasm of hepatocytes, as well as inflammatory infiltration. Again, the administration of LME to HFD-fed mice evidently enhanced the markers of hepatocellular injury evaluated (Figure 6).

### 3.5. Effects of LME on the Systemic Inflammatory Status in Metabolic Tissues

Accumulating evidence suggests that excess nutrient supply promotes lipid storage in adipose tissue by enlarging existing adipocytes (hypertrophy) or forming new ones (hyperplasia) [30,31]. As a result, adipocytes are exposed to a stressful metabolic environment that induces the release of different inflammatory mediators such as interleukin (IL)-6, IL-1β and tumor necrosis factor (TNF)-α, thus facilitating the onset of an oxidative and pro-inflammatory state. In this scenario, several signaling cascades, including those linked to c-jun N-terminal kinases (JNKs), amplify and perpetuate the inflammatory responses in other tissues (i.e., liver), with the subsequent development of metabolic dysfunctions such as insulin resistance and glucose intolerance [32]. As expected, the weights of epididymal and abdominal fat deposits were substantially increased in HFD-fed mice compared to lean ones (Figure 7A). Accordingly, the histological assessment of epididymal fat tissue from untreated HFD-fed mice displayed higher adipocyte expansion when compared with CD-fed mice (Figure 7B). This obese phenotype was intimately related to a higher degree of inflammation, characterized by an increased mRNA expression of several pro-inflammatory mediators, including *Il-1β*, *Il-6*, *Tnf-α*, and *Jnk-1* both in adipose tissue and liver (Figure 7C,D), which contribute to the systemic metabolic dysfunctions during obesity and its related complications. Interestingly, LME treatment markedly reduced these fat deposits and the size of the adipocytes in comparison with control obese mice, which was associated with the amelioration of the systemic inflammatory response (Figure 7A). This beneficial effect displayed by the extract on the inflammatory response can explain the observed enhancement of lipid and glucose metabolism in obese mice since proinflammatory mediators are strongly related to the increment of fatty acid oxidation, lipolysis, as well as insulin resistance [32]. 

It is important to consider that all these biological processes, including energy balance, inflammation, lipid and glucose metabolism, insulin resistance, and adipogenesis, share common ligand-activated transcription factors known as peroxisome proliferator-activated receptors (PPARs), which can be effective in the regulation of obesity-related phenotypes [33]. HFD consumption (and excessive energy storage in fat) alters the expression of PPARα and PPARγ in the adipose tissue, as reported in the present study (Figure 8A). PPARα may influence adipose tissue function, including its inflammatory status [33]. Indeed, it has been described as a modulator of different pathways, including the inhibition of inflammatory genes and the decrease of adipocyte hypertrophy. Similarly, PPARγ, the master regulator of adipogenesis, was reported to reverse macrophage infiltration and upregulate the expression of adiponectin, an abundant peptide secreted by adipocytes [34,35]. Interestingly, the administration of LME to obese mice exerted a beneficial effect by upregulating both transcriptional elements in fat (Figure 8A), thus improving insulin sensitivity and ameliorating the inflammatory status accompanying obesity. These events probably occur via effects on macrophage infiltration and function. Indeed, although less is known about the molecular mechanisms underlying the PPARα anti-inflammatory effects in adipose tissue, PPARγ has been shown to reduce inflammation in activated fat-resident macrophages by interfering with NF-κB signaling pathways [36]. 

It is well known that in order to regulate energy metabolism and immune function, the adipose tissue secretes a large group of bioactive peptides collectively named adipokines, mainly represented by leptin and adiponectin. Abnormal accumulation and dysfunction of adipose tissue during obesity have been linked to alterations of these adipokines [37]. Leptin is a pro-inflammatory peptide, whose secretion by adipocytes is increased in subjects with augmented adipose tissue mass. Great evidence suggests that obesity is linked to increased levels of leptin and reduced expression of its receptor, which contribute to leptin resistance and the inability to suppress appetite or enhance energy expenditure [38]. Unlike leptin, adiponectin is generally related to anti-inflammatory and antioxidant properties in obesity through the downregulation of the expression and the release of several proinflammatory immune mediators [37], especially TNF-α. As expected, the gene expression of both adipokines, *Lep* and *Adipoq,* were altered in the fat tissue of obese mice, as also described in other diet-induced obesity models [29,39], along with a reduced expression of *Lepr* (Leptin Receptor) (Figure 8B). Interestingly, the expression levels of these markers were ameliorated after LME administration to HFD-fed mice, which correlated with the improvement of insulin resistance and the associated hyperglycemia. 

This metabolic dysfunction is well known to be associated with reduced glucose uptake, mainly due to alterations in insulin-stimulated trafficking of glucose transporters [40]. Under normal conditions, insulin regulates glucose uptake in most metabolically active tissues (i.e., the liver, adipose tissue, or skeletal muscles) to control blood glucose levels. This intriguing process occurs via a signaling cascade involving many enzymes, including the serine/threonine kinase AKT, whose phosphorylation and further activation promote glucose-transporters’ translocation (i.e., GLUT2) to the plasma membrane, where they take up extracellular glucose [40]. When obesity-associated insulin resistance occurs, AKT activity is altered, thus leading to defects in the AKT downstream molecules. Consequently, along with an impaired carbohydrate and lipid metabolism, the hepatic glucose output is suppressed, and the insulin-stimulated glucose transport and metabolism in skeletal muscle and adipocytes are decreased. In the present study, a significant down-regulation in hepatic *Glut2* expression was found in HFD mice compared to CD mice, although no differences were observed in the Akt protein expression (Figure 8C). However, LME considerably increased the phosphorylated Akt protein levels, which increased the *Glut2* expression in the liver. Overall, these effects enhance insulin sensitivity and produce lower blood glucose levels through blood glucose uptake. 

Similarly, during obesity-associated insulin resistance status, insulin fails to suppress lipolysis in peripheral tissues, even when nutrient supply is abundant, thus leading to excessive lipid storage in the liver. This event is often accompanied by the inhibition of the AMP-activated protein kinase (AMPK), an energy-sensing enzyme that regulates metabolic homeostasis [41]. As a result, an abnormal increase in hepatic triglyceride accumulation occurs, which, if persisted over time, can induce the development of deleterious conditions underlying the metabolic syndrome [42], such as the reduction of fatty acid oxidation in adipose and other tissues, the reduction of glucose uptake in skeletal muscles, and the stimulation of hepatic glucose production. Moreover, AMPK alterations may contribute to insulin resistance and hyperglycemia by regulating inflammatory signaling in immune cells such as macrophages [43]. All this evidence makes AMPK an attractive therapeutic target considering that its activity is reduced in tissues such as adipose tissue, liver, and skeletal muscle in experimental obesity and humans [41,44]. Our results agree with this observation, since *Ampk* gene expression significantly decreased in the adipose tissue from HFD-fed mice compared to the non-obese group (Figure 8D), while the extract was able to revert it, both in the liver and adipose tissue, thus increasing glucose transport and ameliorating the impaired metabolic functions. Additionally, several studies have demonstrated that regulating lipid and glucose pathways by inducing AMPK phosphorylation is markedly correlated to Sirtuin 1 (Sirt1) expression. Sirt 1 is a key metabolic sensor that directly links environmental nutrient signals to animal metabolic homeostasis. In this sense, excessive lipid accumulation in the liver and adipose tissue can suppress the activity of Sirt1 and reduce the energy regulation capacity of AMPK [45]. In our study, a western blot analysis of hepatic samples from HFD-fed mice showed alteration in these pathways, while the extract was able to revert them. It is worth noting that earlier studies have revealed the potential role of maslinic acid, the main component of LME, in regulating lipogenesis in hepatocytes from obese animals. In fact, maslinic acid reduced liver lipid accumulation by modulating the Sirt1/AMPK signaling pathway [46], in accordance with our findings. Indeed, LME-treated mice showed increased hepatic *Ampk* expression and activation (phosphorylated isoform). Similarly, the treatment with the extract resulted in augmented Sirt1 protein expression in the liver (Figure 8D). This suggests that the high content of maslinic acid in LME could promote the activation and signaling of hepatic Ampk, thus further reducing fat accumulation and steatosis.

### 3.6. Effects of LME Treatment on Intestinal Barrier Dysfunction

It is well documented a that there is a close correlation between obesity and changes in intestinal structure, which can impact gut permeability and result in metabolic complications [47,48]. The present study confirms this since epithelial barrier dysfunction was observed in control HDF-fed mice, evidenced by the reduced expression of several colonic markers of gut integrity, including the peptides Trefoil Factor 3 (*Tff-3*), occludin and tight junction, as well as the mucins (Figure 9A). *Tff-3*, which is expressed by goblet cells and typically co-secreted together with epithelial membrane-bound mucins, such as Muc-1, Muc-2, and Muc-3, to protect the mucus layer [49]. In addition, tight junction protein-1 (Tjp1) works as a tight junction adaptor protein that regulates adherent junctions along with the transmembrane protein occludin, playing a crucial role in maintaining and facilitating epithelial integrity [50]. This altered epithelial integrity was associated with an increase in plasma LPS levels in untreated obese mice, thus promoting a situation of endotoxemia (Figure 9B). 

Of note, previous studies have reported the upregulated expression of these markers of epithelial gut integrity in HFD-fed mice after treatment with different polyphenol-enriched plant extracts [17,28,51]. Similarly, the administration of LME to HFD-fed mice improved the expression of most of these proteins and reduced plasmatic LPS levels (Figure 9B), thus indicating an improvement in the epithelial barrier function and permeability in comparison with obese non-treated mice. Accordingly, several reports have highlighted the strict association between increased LPS plasma levels and dysregulation of the toll-like receptor (TLR)-4 signaling pathway, including increased *Tlr4* expression, which induces pro-inflammatory responses in obesity [52]. Our research confirmed these findings since the hepatic expression of *Tlr4* markedly increased in obese animals compared to the lean ones (Figure 9B). The administration of LME improved the expression of this receptor (Figure 9B), thus confirming an amelioration of the endotoxemia-mediated inflammation in treated-obese mice.

## 4. Conclusions

LME displays a positive impact on decreasing body weight gain and controlling glucose homeostasis in HFD mice. In addition, LME shows antioxidant and adipogenesis inhibitory activities. LME in vivo beneficial effects may be mediated, at least partly, by its antioxidant and anti-inflammatory activities, which are probably ascribed to the synergic properties of its different phytochemical constituents. In sum, our results suggest that LME may be considered a promising complementary approach for the management of obesity and its metabolic complications.

## Figures and Tables

**Figure 1 antioxidants-12-00832-f001:**
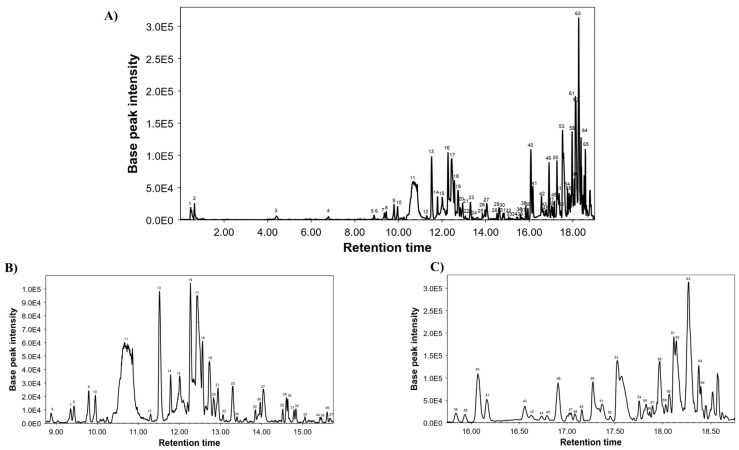
Base peak chromatogram of LME by UHPL-QTOF: (**A**) full chromatogram, (**B**) retention time from 8.50 to 16 min, and (**C**) retention time from 16 to 18.7 min.

**Figure 2 antioxidants-12-00832-f002:**
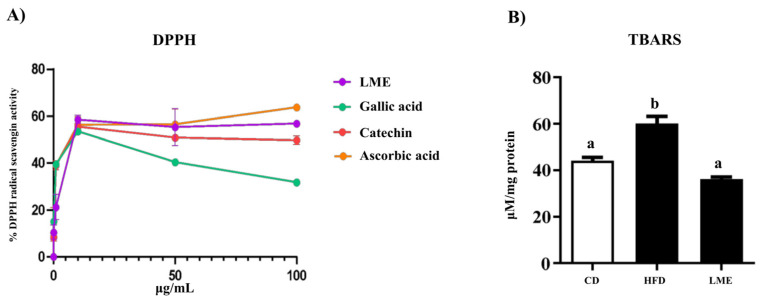
(**A**) DPPH activity scavenging of LME, gallic acid, epicatechin, and ascorbic acid; (**B**) TBARS production in liver lysates. Data are expressed as means ± SEM (*n* = 4). Groups with different letters statistically differ (*p* < 0.05).

**Figure 3 antioxidants-12-00832-f003:**
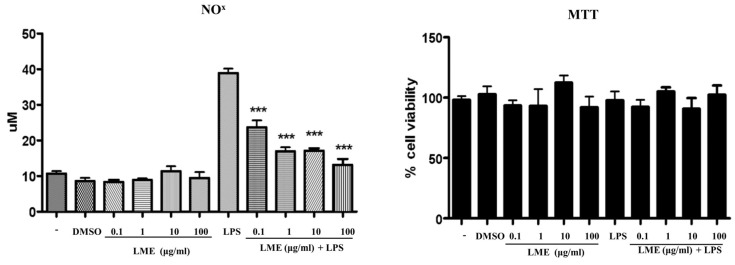
Effects of LME (0.1–100 μg/mL) on nitrite accumulation and cell viability in RAW 264 cells. Data are expressed as means ± SEM. The experiments were performed three times. *** *p* < 0.001 vs. LPS-stimulated cells.

**Figure 4 antioxidants-12-00832-f004:**
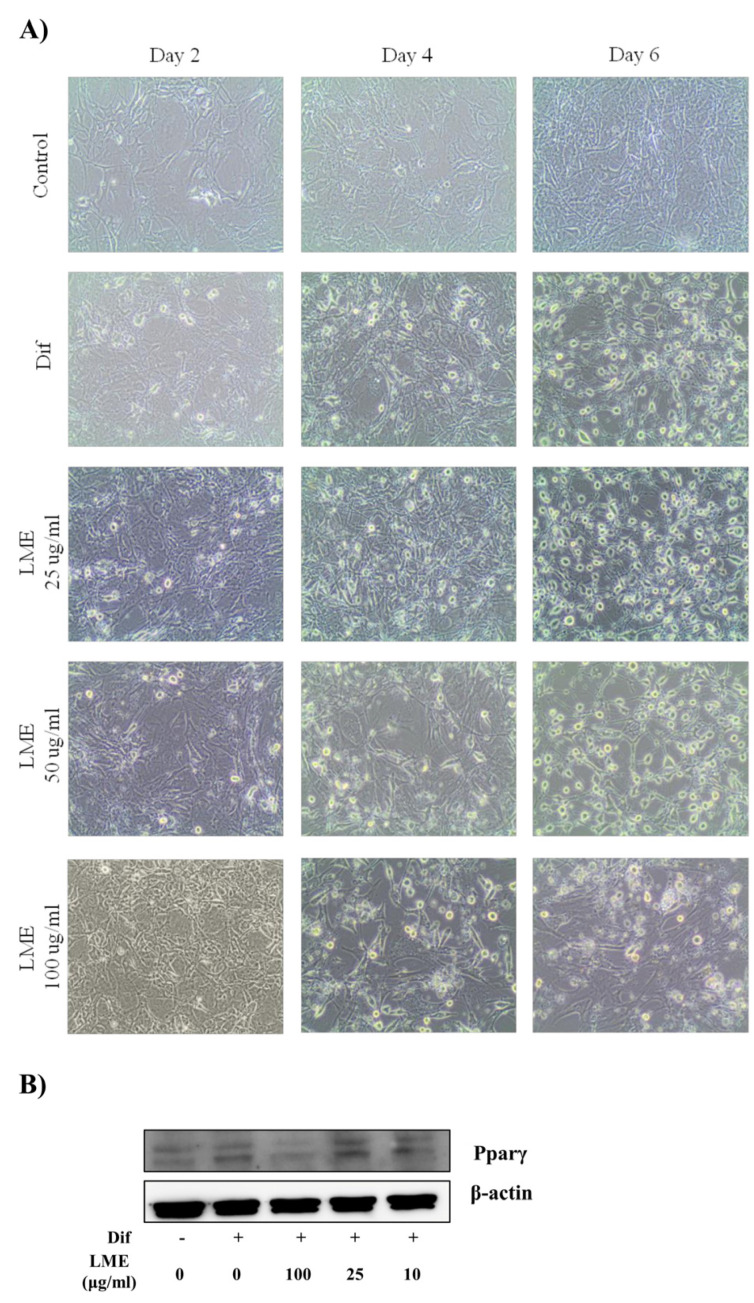
(**A**) Effects of LME (10–100 μg/mL) on adipogenesis (bright cells indicate the presence of lipid accumulation) as well as (**B**) PPARγ protein levels in 3T3-L1 cells. The experiments were performed three times.

**Figure 5 antioxidants-12-00832-f005:**
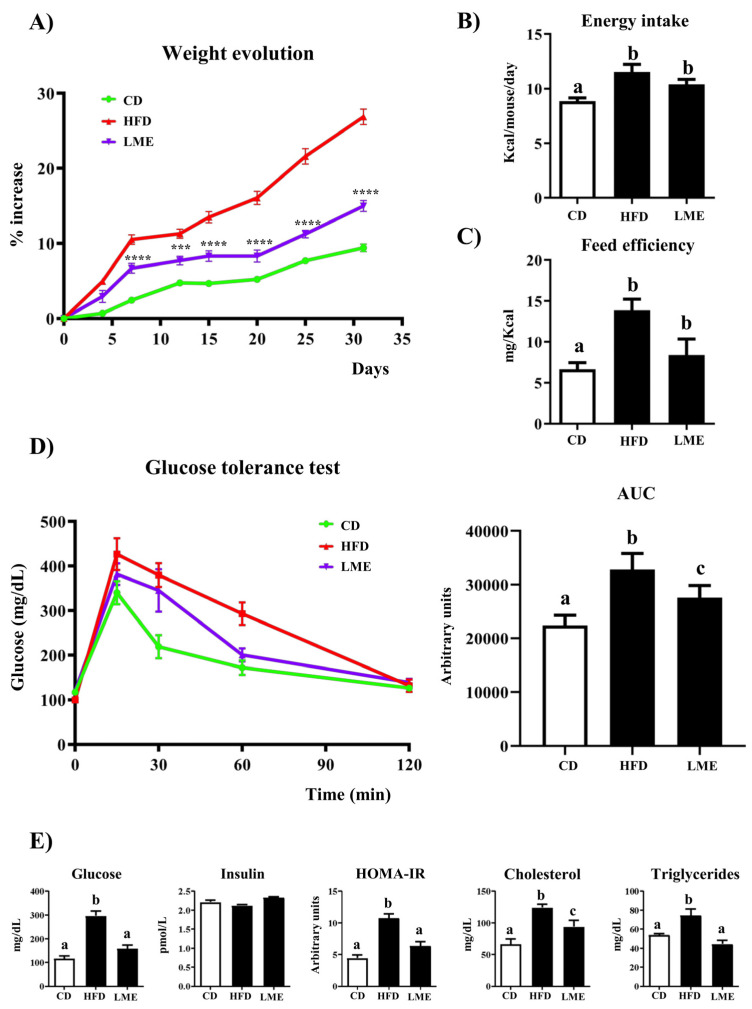
Effects of LME supplementation on (**A**) body weight evolution; (**B**) energy efficiency and (**C**) energy intake; (**D**) glucose tolerance test and area under the curve (AUC); and (**E**) glucose, insulin levels, and HOMA-IR index; total cholesterol plasma levels and triglycerides in control (CD) and high-fat diet (HFD)-fed mice. Data are expressed as means ± SEM (*n* = 8). Groups with different letters statistically differ (*p* < 0.05); *** *p* < 0.001 and **** *p* < 0.0001 vs. HFD-fed mice.

**Figure 6 antioxidants-12-00832-f006:**
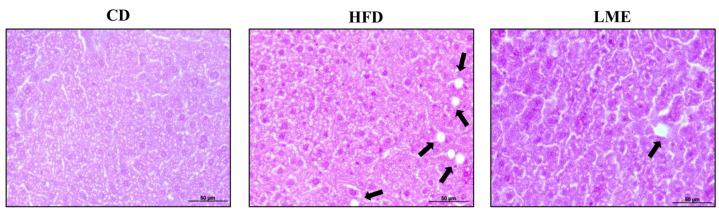
Effects of LME administration on fat deposits in liver tissue stained with hematoxylin and eosin. The black arrows indicate the presence of lipid vacuoles in the cytoplasm of hepatocytes in high-fat diet (HFD)-fed mice.

**Figure 7 antioxidants-12-00832-f007:**
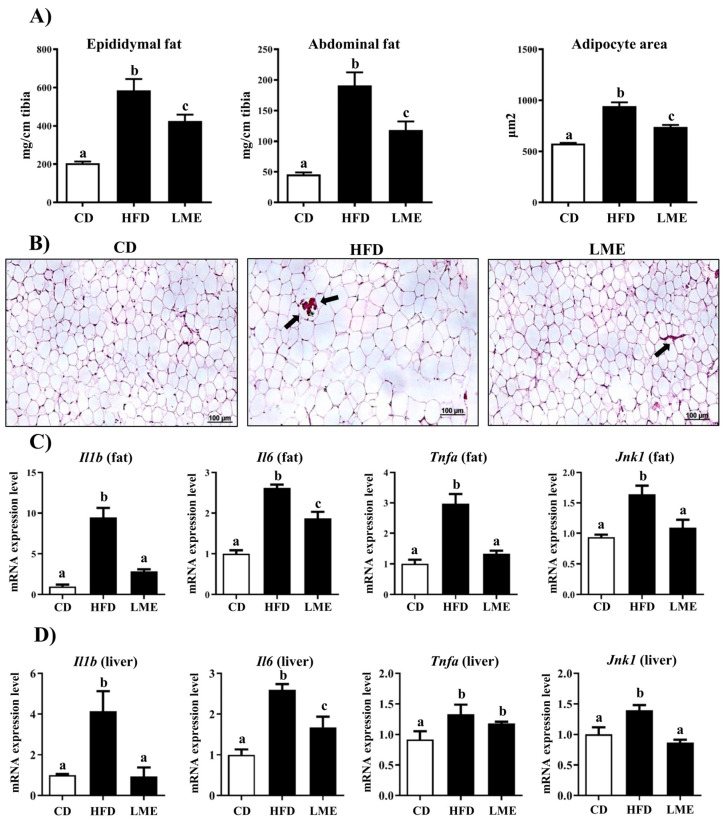
Effects of LME supplementation on (**A**) fat deposits weights; (**B**) epididymal adipose tissue, analyzed by hematoxylin and eosin staining (black arrows show mononuclear inflammatory aggregation); and (**C**,**D**) gene expression of *IL-1β*, *Il-6*, *Tnf-α,* and *Jnk1* in fat and liver. Data are expressed as means ± SEM (*n* = 8). Groups with different letters statistically differ (*p* < 0.05).

**Figure 8 antioxidants-12-00832-f008:**
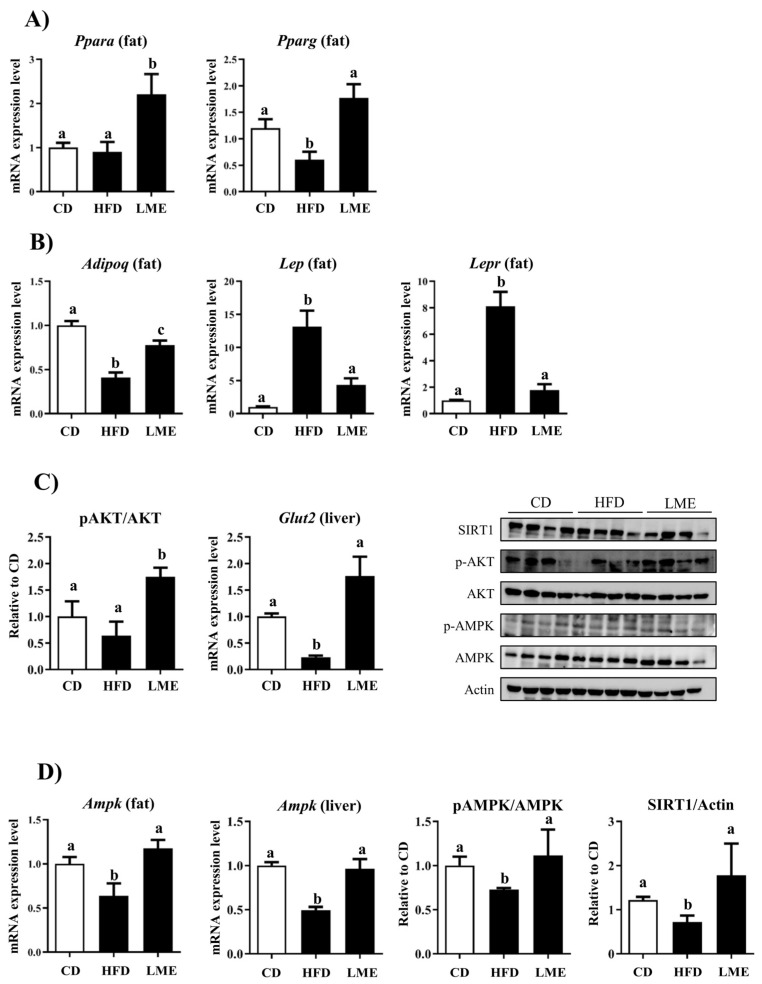
Effects of LME supplementation on fat and liver gene expression of (**A**) *Pparα* and *Pparγ*, (**B**) *Adopoq*, *Lep,* and *Lepr*, as well as (**C**) *Glut2*, pAKT/AKT ratio, (**D**) *Ampk* gene expression, pAMPK/AMPK and SIRT1 protein levels in control (CD), and high-fat diet (HFD)-fed mice, analyzed using real-time qPCR or western blot. Data are expressed as means ± SEM (*n* = 8). Groups with different letters statistically differ (*p* < 0.05).

**Figure 9 antioxidants-12-00832-f009:**
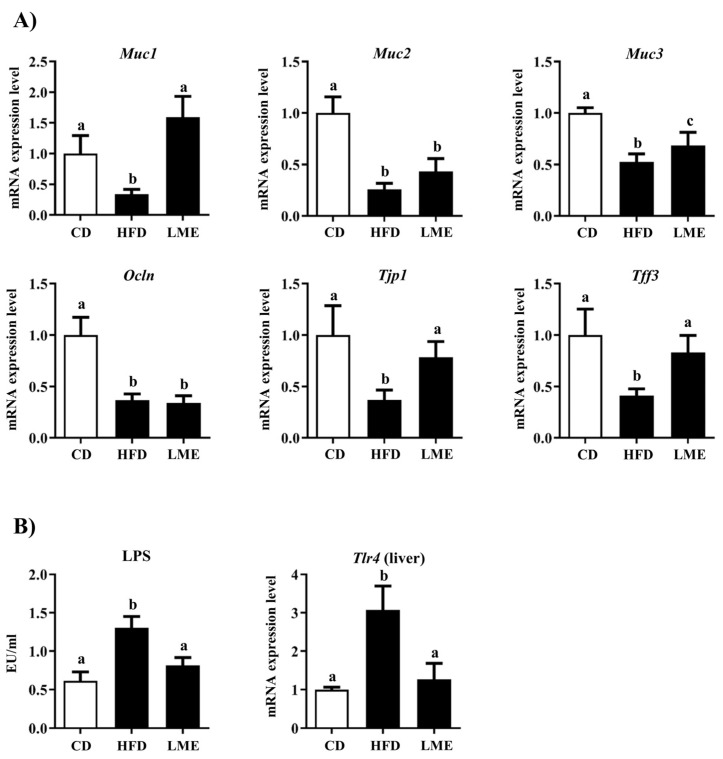
Effects of LME supplementation on: (**A**) markers of intestinal barrier integrity *Muc1*, *Muc2*, *Muc3*, *Occludin*, *Tjp1,* and *Tff3*, as well as (**B**) plasma LPS levels and gene expression of *Tlr4* in the liver. Data are expressed as means ± SEM (*n* = 8). Groups with different letters statistically differ (*p* < 0.05).

**Table 1 antioxidants-12-00832-t001:** Polar profile of LME.

Peak	RT	*m*/*z*	Molecular Formula	Proposed Compounds
1	0.47	343.0356	C_13_H_12_O_11_	Mucic acid lactone gallate
2	0.61	341.1075	C_12_H_22_O_11_	Sucrose
3	4.42	133.0283	C_4_H_6_O_5_	Malic acid
4	6.79	371.0964	C_16_H_20_O_10_	Dihydroferulic acid glucuronide
5	8.87	567.0776	C_26_H_32_O_14_	Phloretin xyloglucoside
6	8.89	301.0705	C_16_H_14_O_6_	Hesperetin
7	9.35	463.0876	C_21_H_20_O_12_	Quercetin glucoside
8	9.43	593.0958	C_29_H_22_O_14_	(Epi)catechin digallate
9	9.79	447.0922	C_21_H_20_O_11_	Luteolin 7-*O*-glucoside
10	9.96	609.1448	C_27_H_30_O_16_	Rutin
11	10.71	473.0711	C_22_H_18_O_12_	Chicoric acid
12	11.3	489.1024	C_23_H_22_O_12_	Kaempferol acetyl-glucopyranoside
13	11.52	477.0664	C_21_H_18_O_13_	Quercetin glucuronide
14	11.79	461.0717	C_21_H_18_O_12_	Isoscutellarin 8-*O*-glucoronide
15	12.01	503.3370	C_30_H_48_O_6_	Madecassic acid or its isomer
16	12.27	491.0823	C_22_H_20_O_13_	Isorhamnetin 3-*O*-glucuronide
17	12.44	307.0446	C_14_H_12_O_8_	Fulvic acid analogue 1
18	12.57	839.4089	C_42_H_64_O_17_	Yunganoside G2 or its isomer
19	12.74	519.0928	C_27_H_20_O_11_	Citreaglycon A
20	12.83	533.1661	C_26_H_30_O_12_	Amurensin
21	12.94	839.4052	C_42_H_64_O_17_	Yunganoside G2 or its isomer
22	13.07	545.3464	C_32_H_50_O_7_	Hovenidulcigenin B or its isomer
23	13.29	839.4059	C_42_H_64_O_17_	Yunganoside G2 or its isomer
24	13.41	545.3454	C_32_H_50_O_7_	Hovenidulcigenin B or its isomer
25	13.86	939.3139	C_50_H_52_O_18_	Unknown
26	13.89	327.2169	C_18_H_32_O_5_	Fatty acid
27	14.07	307.0446	C_14_H_12_O_8_	Fulvic acid analogue 2
28	14.52	503.3365	C_30_H_48_O_6_	Madecassic acid or its isomer
29	14.60	823.4134	C_42_H_64_O_16_	Licoricesaponin J2 or its isomer
30	14.64	329.2487	C_30_H_48_O_7_	Fatty acid
31	14.79	823.4134	C_42_H_64_O_16_	Licoricesaponin J2 or its isomer
32	14.82	519.3374	C_30_H_48_O_7_	Hydroxyecdysone monoacetonide
33	15.06	287.2228	C_16_H_32_O_4_	Fatty acid
34	15.43	501.3208	C_30_H_46_O_6_	Medicagenic acid or its isomer
35	15.46	777.2611	C_41_H_46_O_15_	Guaiacylglycerol buddlenol A
36	15.6	501.3208	C_30_H_46_O_6_	Medicagenic acid or its isomer
37	15.66	501.3304	C_30_H_46_O_6_	Medicagenic acid or its isomer
38	15.84	503.3363	C_30_H_48_O_6_	Madecassic acid or its isomer
39	15.93	503.3361	C_30_H_48_O_6_	Madecassic acid or its isomer
40	16.07	503.3358	C_30_H_48_O_6_	Madecassic acid or its isomer
41	16.16	503.3359	C_30_H_48_O_6_	Madecassic acid or its isomer
42	16.56	503.3352	C_30_H_48_O_6_	Madecassic acid or its isomer
43	16.63	503.3351	C_30_H_48_O_6_	Madecassic acid or its isomer
44	16.73	503.3347	C_30_H_48_O_6_	Madecassic acid or its isomer
45	16.76	677.3508	C_36_H_54_O_12_	Bryoamaride or its isomer
46	16.9	677.353	C_36_H_54_O_12_	Bryoamaride or its isomer
47	17.04	485.3261	C_30_H_46_O_5_	Quillaic acid or its isomer
48	17.08	485.3249	C_30_H_46_O_5_	Quillaic acid or its isomer
49	17.16	441.3369	C_29_H_46_O_3_	Camellenodiol
50	17.24	487.3403	C_30_H_48_O_5_	Asiatic acid or its isomer
51	17.37	677.3522	C_36_H_54_O_12_	Bryoamaride or its isomer
52	17.45	487.3406	C_30_H_48_O_5_	Asiatic acid or its isomer
53	17.52	487.3409	C_30_H_48_O_5_	Asiatic acid or its isomer
54	17.75	295.2265	C_18_H_32_O_3_	Fatty acid
55	17.81	471.3475	C_30_H_48_O_4_	Maslinic acid or its isomer
56	17.86	293.2109	C_18_H_30_O_3_	Fatty acid
57	17.89	425.3413	C_29_H_46_O_2_	Stigmastene dione
58	17.96	469.3408	C_30_H_46_O_4_	Glycyrrhetinic acid
59	18.03	471.3473	C_30_H_48_O_4_	Maslinic acid or its isomer
60	18.06	471.3464	C_30_H_48_O_4_	Maslinic acid or its isomer
61	18.11	471.347	C_30_H_48_O_4_	Maslinic acid or its isomer
62	18.14	471.3461	C_30_H_48_O_4_	Maslinic acid or its isomer
63	18.27	471.3476	C_30_H_48_O_4_	Maslinic acid or its isomer
64	18.38	277.2156	C_18_H_30_O_2_	Fatty acid
65	18.4	467.3159	C_27_H_48_O_6_	Fatty acid

**Table 2 antioxidants-12-00832-t002:** Total phenolic content and antioxidant capacity of LME.

Method	Value
Folin-Ciocalteu (mg GAE/g d.e.)	179 ± 1
FRAP (mmol eq. FeSO_4_/g d.e.)	2.576 ± 0.002
TEAC (mmol eq. Trolox/g d.e.)	1.30 ± 0.02
ORAC (mmol eq. Trolox/g d.e.)	2.08 ± 0.09

GAE—gallic acid equivalents; d.e.—dry extract; eq.—equivalents.

## Data Availability

The data presented in this study are available on request from the corresponding author.

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
