# Peer review of "The Antioxidant Properties of Lavandula multifida Extract Contribute to Its Beneficial Effects in High-Fat Diet-Induced Obesity in Mice"

_antioxidants, 2023, doi:10.3390/antiox12040832_

Round 1
Reviewer 1 Report
Manuscript No Antioxidants-2288969
„The Antioxidant Properties of Lavandula multifida Extract Contribute to Its Beneficial Effects in High Fat Diet-induced Obesity in Mice” for Antioxidants
Comments:
1. Please remove the dot after the title of the manuscript.
2. Materials and methods. 2.4. Line 131. Please change the comma to a dot when describing fractional numerical values.
3. Please check the correctness of superscripts and subscripts in the text.
4. Materials and methods. 2.4. I understand that primary antibodies are rabbit anti-mouse. However, please indicate this in the text, since secondary antibodies are described by their origin.
5. Materials and methods. 2.5. Line 151. Please check the concentration of the applied streptomycin. Typically, the concentration in in vitro cell cultures is 100 microg/mL.
6. Figure 4. Please indicate in the figure caption what effect it is and onto images indicate what the reader should pay attention to. Similarly figures 5B and 9.
Author Response
REVIEWER 1
Manuscript No Antioxidants-2288969 „The Antioxidant Properties of Lavandula multifida Extract Contribute to Its Beneficial Effects in High Fat Diet-induced Obesity in Mice” for Antioxidants
Comments:
- Please remove the dot after the title of the manuscript.
Response: Following the reviewer suggestion, the dot has been removed.
- Materials and methods. 2.4. Line 131. Please change the comma to a dot when describing fractional numerical values.
Response: Accordingly, the comma has been replaced by a dot.
- Please check the correctness of superscripts and subscripts in the text.
Response: The manuscript has been revised.
-Materials and methods. 2.4. I understand that primary antibodies are rabbit anti-mouse. However, please indicate this in the text, since secondary antibodies are described by their origin.
Response: The antibody information has been revised and modified.
-Materials and methods. 2.5. Line 151. Please check the concentration of the applied streptomycin. Typically, the concentration in in vitro cell cultures is 100 microg/mL.
Response: We apologise for the mistake.
-Figure 4. Please indicate in the figure caption what effect it is and onto images indicate what the reader should pay attention to. Similarly figures 5B and 9.
Response: According to the suggestion made by the reviewer, more detailed information has been included in the figure caption of the revised manuscript.
Reviewer 2 Report
The article entitled " The Antioxidant Properties of Lavandula multifida Extract Contribute to Its Beneficial Effects in High Fat Diet-induced Obesity in Mice" presented to me for review is original work. The authors try to prove the effects of a characterized hydroalcoholic extract of L. multifida (LME) on an experimental model of diet-induced obesity in mice, paying special attention to its impact on the altered metabolic and inflammatory status. In my opinion, the article is interesting and raises the important problem of obesity, which is currently one of the most important problems in the world because it causes the development of many diseases. Starting from the beginning, the abstract is a bit uninformative and quite laconic. This is an important element for the reader, so it should contain all the necessary research results in a fairly condensed form - please rewrite. Please remove the dot at the end of the article title. The introduction, on the other hand, is short but contains all the important things that define the whole article. The methodology section is described quite correctly, but please move the table with genes as supplementary material because it is too chaotic in this place and does not bring any outstanding information. Also, I think authors should add a product length column. Also, I have one question for the authors regarding the genes used in the research. Did the reaction yield in such large amounts of products as, for example, over 500 for GAPDH gene, not affect the reaction yield?? For this type of analysis, you should rather choose a product genes sizes of max 150 bp. Additionally, in section 2.11 please describe in more detail the amounts of individual components used in the reaction. The results combined with the discussion are a bit too heavy for the reader because they do not allow you to focus on the ,,clou’’ of the topic. In addition, the authors poorly discuss their results with the literature, omitting citations directly related to the functioning mechanisms and activity of genes or proteins. I suggest that the authors check a little more whether this research plant actually does not have the studies described by the authors. For example, section 3.3 has no literature cited and 3.4 (first page) is described with a single reference 26.- I think that's a bit too weak, and the results should be directly confronted with the work of other researchers. I would suggest the authors to either rewrite this section or split it into two separate ones, i.e. results and discussion, which will certainly become more transparent to the reader. Please correct the chromatogram (3.1 section) in the article because in this form it is difficult to read and mark the individual numbers on it corresponding to the specific analyzed compounds, it is quite important because the authors only marked the compounds qualitatively and not quantitatively. Besides, in the results, please correct the descriptions of the figures for gene expression, not as "fold increase" but rather mRNA expression level". The conclusion section is very clear and summarizes all the authors' achievements. The work requires re-reading and correcting some typos and punctuation errors.
Author Response
The article entitled " The Antioxidant Properties of Lavandula multifida Extract Contribute to Its Beneficial Effects in High Fat Diet-induced Obesity in Mice" presented to me for review is original work. The authors try to prove the effects of a characterized hydroalcoholic extract of L. multifida (LME) on an experimental model of diet-induced obesity in mice, paying special attention to its impact on the altered metabolic and inflammatory status. In my opinion, the article is interesting and raises the important problem of obesity, which is currently one of the most important problems in the world because it causes the development of many diseases. Starting from the beginning, the abstract is a bit uninformative and quite laconic. This is an important element for the reader, so it should contain all the necessary research results in a fairly condensed form - please rewrite. Please remove the dot at the end of the article title.
Response: Following the reviewer suggestion, the abstract and the title have been modified.
The introduction, on the other hand, is short but contains all the important things that define the whole article. The methodology section is described quite correctly, but please move the table with genes as supplementary material because it is too chaotic in this place and does not bring any outstanding information. Also, I think authors should add a product length column. Also, I have one question for the authors regarding the genes used in the research. Did the reaction yield in such large amounts of products as, for example, over 500 for GAPDH gene, not affect the reaction yield?? For this type of analysis, you should rather choose a product genes sizes of max 150 bp. Additionally, in section 2.11 please describe in more detail the amounts of individual components used in the reaction. The results combined with the discussion are a bit too heavy for the reader because they do not allow you to focus on the ,,clou’’ of the topic. In addition, the authors poorly discuss their results with the literature, omitting citations directly related to the functioning mechanisms and activity of genes or proteins. I suggest that the authors check a little more whether this research plant actually does not have the studies described by the authors. For example, section 3.3 has no literature cited and 3.4 (first page) is described with a single reference 26.- I think that's a bit too weak, and the results should be directly confronted with the work of other researchers. I would suggest the authors to either rewrite this section or split it into two separate ones, i.e. results and discussion, which will certainly become more transparent to the reader. Please correct the chromatogram (3.1 section) in the article because in this form it is difficult to read and mark the individual numbers on it corresponding to the specific analyzed compounds, it is quite important because the authors only marked the compounds qualitatively and not quantitatively. Besides, in the results, please correct the descriptions of the figures for gene expression, not as "fold increase" but rather mRNA expression level". The conclusion section is very clear and summarizes all the authors' achievements. The work requires re-reading and correcting some typos and punctuation errors.
Response: We thank the reviewer for the valuable comments. We have rewritten and restructured the results & discussion section to make it more clear and easy to follow. Moreover, we have included more references to previous works to discuss the results.
Regarding the use of the GAPDH primers, we have validated them in the same settings, using other housekeeping genes with smaller amplicons, and we have observed no differences. According to the suggestion made by the reviewer, we have moved the table with specific primers (Table S1) to supplementary material.
We have also modified the chromatogram and corrected the description of the figures of gene expression as suggested by the reviewer.
Reviewer 3 Report
Dear authors,
Let me start by congratulating you on the article.
Overall, the manuscript is complex and contain interesting information’s. The abstract is appropriate and the aim of the work clearly established. Regarding the methodology used, I appreciated first of all the very clear structure and the multitude of tests used. The results are presented in a clear manner. Chapter of discussion is drawn up in a clear and logical manner, making reference to the relatively recent literature data.
There are some negative aspects, which I believe need to be clarified and corrected in order to publish this manuscript.
- The abstract is much too general. It is necessary to enter some precise information concerning the results. A short presentation (or a short list) of the parameters analyzed must be presentend
- Regarding the methodology:
Line 124 – “The aerial parts Folin-Ciocalteu (TPC), FRAP, TEAC and ORAC were performed on....” please rephrase, it is not very clear
Lines 124 – 143: Why did you decide to only describe the DPPH technique? You adjusted the described technique in some way? Providing some information about the methodology that was used for each of the analyzed parameters is preferable.
Presentation of the results obtained in the in vitro study: 3.6. Effects of LME on nitrite production in RAW 264 cells and adipogenesis in 3T3-L1 cells: I think it would make more sense and be more suitable to present it before the in vivo results.
Author Response
Dear authors, let me start by congratulating you on the article. Overall, the manuscript is complex and contains interesting information. The abstract is appropriate and the aim of the work clearly established. Regarding the methodology used, I appreciated first of all the very clear structure and the multitude of tests used. The results are presented in a clear manner. Chapter of discussion is drawn up in a clear and logical manner, making reference to the relatively recent literature data.
There are some negative aspects, which I believe need to be clarified and corrected in order to publish this manuscript.
- The abstract is much too general. It is necessary to enter some precise information concerning the results. A short presentation (or a short list) of the parameters analyzed must be presented.
Response: Thank you for your suggestion, which echoes one of the main comments of Reviewer 2. The abstract has been modified, thus including detailed information concerning the results and a short list of the parameters analyzed.
- Regarding the methodology:
Line 124 – “The aerial parts Folin-Ciocalteu (TPC), FRAP, TEAC and ORAC were performed on....” please rephrase, it is not very clear
Response: The text has been modified to make it more clear.
Lines 124 – 143: Why did you decide to only describe the DPPH technique? You adjusted the described technique in some way? Providing some information about the methodology that was used for each of the analyzed parameters is preferable.
Response: We decided to describe the method because in the reference we used it was not described in detail. However, the other methods are extensively explained in the references.
Presentation of the results obtained in the in vitro study: 3.6. Effects of LME on nitrite production in RAW 264 cells and adipogenesis in 3T3-L1 cells: I think it would make more sense and be more suitable to present it before the in vivo results.
Response: As suggested by the reviewer, the results section has been restructured.
Round 2
Reviewer 2 Report
Please take a look because there are some punctuation and stylistic errors in the work after corrections.
Author Response
Following the editor and reviewer suggestions, we have revised the manuscript and made the corresponding changes. Moreover, we thank the reviewer for the thorough and complete revision. Consequently, punctuation and stylistic errors have been corrected and the manuscript has been double checked.